# Association between Temporal Glycemic Change and Risk of Pancreatic Cancer in Men: A Prospective Cohort Study

**DOI:** 10.3390/cancers14143403

**Published:** 2022-07-13

**Authors:** Jie Cai, Hongda Chen, Ming Lu, Yuhan Zhang, Bin Lu, Chenyu Luo, Xiaoshuang Feng, Lei You, Min Dai, Yupei Zhao

**Affiliations:** 1Department of General Surgery, Peking Union Medical College Hospital, Chinese Academy of Medical Sciences and Peking Union Medical College, Beijing 100730, China; caijie1113@126.com (J.C.); florayo@163.com (L.Y.); 2Medical Research Center, Peking Union Medical College Hospital, Chinese Academy of Medical Sciences and Peking Union Medical College, Beijing 100730, China; chenhongda2005@163.com (H.C.); luming2018@hotmail.com (M.L.); zyhmsf426@163.com (Y.Z.); lb838744529@126.com (B.L.); luochy23@163.com (C.L.); 3Genomic Epidemiology Branch, International Agency for Research on Cancer, 69372 Lyon, France; fengx@iarc.fr

**Keywords:** blood glucose, variation, pancreatic cancer, cohort study, risk factor

## Abstract

**Simple Summary:**

Hyperglycemia has been reported to increase the risk of pancreatic cancer, while the association between temporal glycemic changes and the risk of pancreatic cancer remains uncertain. This study used data from a prospective cohort and found a U-shaped association between annual change in fasting blood glucose and risk of pancreatic cancer in males. A higher coefficient of variance and a greater range of fasting blood glucose were also associated with the increased risk of pancreatic cancer. These findings may help identify individuals at high risk for pancreatic cancer; thus, appropriate measures can be taken to prevent pancreatic cancer.

**Abstract:**

Hyperglycemia has been reported to increase the risk of pancreatic cancer (PC), while the association between glycemic change and PC risk has rarely been explored. Using data from a prospective cohort study conducted in China since 2006, 138,870 males with available fasting blood glucose (FBG) levels, including 106,632 males with at least two FBG measurements, were analyzed. The associations between FBG (level, change, and stability) and PC incidence were evaluated using Cox proportional hazard regression and restricted cubic splines. Baseline (*p* = 0.109) and recent (*p* = 0.070) FBG levels and incident PC were not significantly associated. U-shaped associations were observed between the annual FBG change and PC risk. Compared with stable FBG, participants with annual FBG change rates <−0.05 mmol/L or >0.15 mmol/L had about four-fold (HR, 4.010; 95% CI: 1.920–8.375) and six-fold (HR, 5.897; 95% CI: 2.935–11.848) higher PC risks, respectively. The PC risk increased by 2.5% (HR_linear_ = 1.025, 95% CI:1.009–1.042) for every 1% increase in the coefficient of variation for FBG. A subgroup analysis of males without diabetes at baseline showed stronger associations. Temporal FBG changes may be an important factor for identifying populations with high PC risks.

## 1. Introduction

Pancreatic cancer (PC), the seventh most lethal malignant tumor in the world [1], has shown an increasing global trend both in its incidence and mortality in past decades [2]. Patients in the distant stage had a 5-year relative survival rate of 3.1%, while, in patients in the localized stage, this rate could reach 43.9% [3], indicating the importance of the prevention and early detection of PC. In addition to non-modifiable risk factors associated with PC development, such as age, sex, and genetic susceptibility, modifiable risk factors mainly include obesity, type 2 diabetes, and tobacco use [4,5,6]. According to current established guidelines [7,8], PC screening is not recommended for the general population. Consequently, early detection programs mainly focus on high-risk individuals, such as patients with a strong family history or genetic predisposition, and those with pancreatic cystic lesions, although these patients represent only 15–20% of those presenting with PC. In some recent publications, type 2 diabetes, particularly new-onset diabetes, has been used as a new predictor in PC risk models to help identify high-risk groups [9,10,11].

As a quantitative indicator of diabetes, increased fasting blood glucose (FBG) has also been reported to be associated with a higher risk of PC [12,13]. A meta-analysis showed that every 0.56 mmol/L (10 mg/dL) increase in FBG level was associated with a 14% increased risk of PC (95%CI, 1.06–1.22) [14]. However, current studies on the association between FBG and PC are mostly concentrated on a single or average level, and studies on the association between the FGB change and PC are limited.

In fact, FBG levels change dynamically. These changes and stability can better reflect the individual’s metabolic state. Several cross-sectional studies have observed progressive hyperglycemia before PC diagnosis [15,16], indicating that previous changes in FBG levels may be associated with PC risk. However, these studies did not quantify the FBG changes specifically and provided limited evidence due to their retrospective study design and small sample sizes. Therefore, further evidence for the association between FBG change and the risk of PC is still needed. 

In this study, we used the prospective Kailuan cohort [17,18] to further explore the association between the changes in or stability of FBG, and the risk of PC. We anticipate that the findings of our study may contribute to a more accurate identification of populations at high risk of PC and may provide timely preventative measures to reduce the overall burden of this disease.

## 2. Materials and Methods

### 2.1. Study Population and Design

The present study was conducted using the Kailuan cohort, the cohort from a study conducted since 2006. The data were based on health examinations conducted every two years for employees of Kailuan Group in the city of Tangshan, Hebei Province, northern China. Kailuan Group is famous for its coal industry, but in fact, the group is a comprehensive company with employees in many different fields, such as coal production, machine manufacturing, transportation, chemical production, education, and health care. Eleven hospitals are affiliated with Kailuan Group and are in charge of the health examination of employees. Employees of Kailuan Group who were older than 18 years (retired individuals included) were invited to participate in health examinations, which included questionnaire assessments, clinical examinations, and laboratory tests. Details of the cohort design and procedures have been published previously [17,18]. By the end of 2017, the Kailuan cohort study had conducted six rounds of health examinations and included a total of 175,670 participants. Due to the limited number of women (less than 20%) and female PC cases (14 female PC cases), this study only focused on men.

Participants in the Kailuan cohort who met the following criteria were included in the study: (1) men older than 18 years (2) who had signed informed consent and (3) whose baseline FBG measurements were obtained at recruitment. To avoid potential survival bias, subjects who were diagnosed with any existing cancer before cohort entry were excluded from the study. Participants with severe abnormal FBG levels (>30 mmol/L) were also excluded. Exposure information was collected between January 2006 and December 2017. Finally, 138,870 male participants were included in the analysis. Of these, 106,632 men with at least two FBG measurements were included in the analysis of FBG changes and stability.

### 2.2. Assessment of FBG and FBG Change

Venous blood samples were obtained from participants in the morning, after overnight fasting, to measure FBG concentration, and all tests were conducted in the central laboratory of Kailuan General Hospital using a standard operating procedure. Baseline FBG values, also termed the first FBG, were defined as the FBG measurement obtained when the participant entered the cohort. The last FBG was defined as the last FBG measurement obtained before the diagnosis of PC or by the end of follow-up (31 December 2019). We also investigated the most recent FBG value of participants, which was the last FBG obtained within 2 years before PC diagnosis or at the end of follow-up. Single FBG levels were grouped as normal fasting glucose (NFG: FBG < 5.6 mmol/L), impaired fasting glucose (IFG: 5.6 ≤ FBG < 7.0 mmol/L), and diabetic fasting glucose (DFG: FBG ≥ 7.0 mmol/L), according to the category recommended by the American Diabetes Association [19]. For those with at least two FBG measurements, the temporal FBG change was defined as the last FBG value subtracted from the first FBG value. The annual FGB change was defined as the temporal FBG change divided by the duration of change (time interval between the first FBG and last FBG). These groups were categorized according to the percentage distribution, such as P25, P50, and P75. The index for FBG stability, including the coefficient of variation (CV), range, and arithmetic mean over time, were calculated from all available FBG measurement results for each participant.

### 2.3. Assessment of Covariates

Age at baseline was calculated based on birth date and the time of entry into the cohort. Height and weight were measured in each health examination, and body mass index (BMI) was then calculated as weight in kilograms divided by height in meters squared. For participants with missing BMI (accounting for less than 5%) in the health examination, the median BMI of that health examination round was used as a substitute. Information on cigarette smoking history, alcohol drinking history, education level, and anti-diabetes drug use was also obtained from the questionnaires. We used missing indicators for the missing values of categorical variables.

### 2.4. Outcome Ascertainment

We followed participants from the baseline examination up to the occurrence of any cancer, death, or 31 December 2019, whichever came first. Incident cancer cases were identified by self-reported information in questionnaires administered every two years and by linkage with the Kailuan social security system, Tangshan medical insurance system, and provincial vital statistical data annually until 31 December 2019. Cancer information was reconfirmed by evaluating the records on discharge summaries from the hospitals at which participants were diagnosed and treated. The medical records of cancer cases were further reviewed by clinical experts to confirm the diagnosis time, location, and other information on the tumors. Cancers were coded according to the International Classification of Diseases, Tenth Revision (ICD-10), and PC was coded as C25.

### 2.5. Statistical Analysis

The person-years for participants were calculated from the beginning of the baseline examination until the end of the follow-up. Variables with a non-normal distribution were described as median, 25th percentile, and 75th percentile (continuous variables), or as number and percentage (categorical variables). A non-parametric test was used for comparison between the groups. Cox proportional hazard regression models were used to compute hazard ratios (HRs) and 95% confidence intervals (95% CIs). The proportional hazard assumption was examined by testing for time-by-covariate interactions and Schoenfeld residuals. A restricted cubic spline (RCS) was used to delineate nonlinear associations. Due to the influence of the baseline FBG levels, we further performed a subgroup analysis for patients without diabetes at baseline (FBG < 7 mmol/L, and without anti-diabetes drug-use history) when studying changes in FBG levels.

Sensitivity analyses were conducted (1) by excluding coal-mine workers to remove potential health worker effects; (2) by only including participants > 30 years and < 80 years; (3) by only including participants with normal BMI or overweight (18.5 kg/m^2^ < BMI ≤ 30 kg/m^2^); (4) by excluding patients diagnosed with cancer within 3 years after entering the cohort, to reduce the probability of causality inversion; (5) by excluding participants with >1.5-fold increase from the threshold for white blood cell counts (>15 × 10^9^/L) or C-reactive protein (>15 mg/L) to avoid the potential effects of inflammation on blood glucose metabolism; and (6) by excluding participants who had ever used anti-diabetes drugs.

Two-sided statistical tests were performed, and *p* values < 0.05 were considered statistically significant. All analyses were performed using SAS version 9.4 (SAS Institute Inc., Cary, NC, USA).

## 3. Results

### 3.1. Participant Characteristics

A total of 138,870 men with baseline FBG measurements were included in the analysis (Table 1). Of these, 32,238 had only one FBG measurement and were not included in the analysis to examine the change in and stability of the FBG. There were 44,479 and 62,153 participants who underwent FBG measurements 2–3 times and 4–6 times, respectively. All participants were followed-up for a median of 11 years. The median age of all participants was 51 years, and half of them were between 40 and 60 years old. About 70% of them only finished junior middle school or lower education. Nearly half of the participants had a history of cigarette smoking and alcohol drinking, while less than 2% of participants had an anti-diabetic medication-use history. Participants in the groups with more FBG measurements were younger and had a longer follow-up time and FBG-change measurement duration. Higher rates of smoking and drinking, and lower rates of diabetes drug use were observed in groups with more FBG measurements. In total, 135 cases of PC were diagnosed. The ages of these patients ranged from 46 to 85 years, and the median age was 66 years.

### 3.2. Association of Baseline FBG or the Most Recent FBG with PC

A total of 138,870 participants (1,400,848 person-years) and 22,451 participants (196,106 person-years) were included in the analysis of baseline FBG and recent FBG, respectively. No significant association between hyperglycemia (IFG and DFG) and pancreatic cancer risk was observed, neither in baseline FBG nor in a recent FBG study, when compared with the normal FBG level. The linear HRs for baseline FBG and the most recent FBG were 1.038 (95% CI: 0.992–1.087) and 1.048 (95% CI: 0.996–1.102), respectively (Appendix A). RCS models did not show any non-linear association for either baseline or recent FBG with PC risk (Appendix A).

### 3.3. Association of FBG Change with PC

A total of 106,632 men who had at least two FBG measurements (total person-years of 1,184,216) were included in the analysis of FBG changes. The median duration of FBG change in participants with at least two FBG measurements was 8 years. Among these 106,632 participants, 9136 (12 PC cases) had diabetes at baseline while 97,496 (81 PC cases) did not.

The Cox regression analysis for annual FGB change and PC risk in men is shown in Table 2. When setting the annual FGB change at the range of −0.05 and 0.05 mmol/L as a reference, multivariate analyses with adjustments for potential confounders showed that individuals with FBG that annually increased by more than 0.15 mmol/L had a significantly higher risk of PC, with an HR of 5.897 (95% CI: 2.935–11.848). In contrast, a four-fold increased risk was observed in individuals whose FBG annually decreased by more than 0.05 mmol/L (adjusted HR, 4.010; 95% CI: 1.920–8.375). Similarly, when setting an annual FGB change of between −1% and 1% as the reference, more than three-fold (adjusted HR, 3.851; 95% CI:1.883–7.878) and five-fold (adjusted HR, 5.512; 95% CI:2.827–10.748) risk of PC were observed in individuals with an annual decrease of more than 1% or increase of more than 3% in FBG, respectively.

When restricting the analysis to individuals who did not have diabetes at baseline, we observed a similar pattern, with an adjusted HR of 3.976 (95% CI: 1.797–8.798) and 6.769 (95% CI: 3.254–14.080) for those whose FBG annually decreased more than 0.05 mmol/L or increased more than 0.15 mmol/L, respectively (Table 2). However, when restricting the analysis to individuals with diabetes at baseline, no significant associations were observed (Appendix A), partly due to the limited sample size. In addition to the annual change in FBG levels, total temporal changes in FBG were also significantly associated with PC risk (Appendix A). Sensitivity analyses for associations between annual FGB changes and PC risk showed consistent associations (Appendix A).

RCS showed a significant U-shaped non-linear association between annual FGB change and risk of PC (Figure 1). The risk of PC increased significantly as FBG increased, and the risk stabilized when FBG increased by over 0.2 mmol/L or 4% annually. Interestingly, a significant risk for PC incidence was shown in the decreased annual FBG percentage (Figure 1b) while not shown in the decreased FBG value (Figure 1a), indicating the effects of baseline FBG levels on the associations. In males without baseline diabetes, the trends of associations were consistent in both FBG value and percentage, which showed more stable associations than that in all males.

### 3.4. FBG Stability

Participants with at least two FBG measurements were included in the analysis of FBG stability. The Cox regression analysis showed that a higher CV and greater range of FBG levels during a period were associated with an increased risk of PC (Table 3). When setting an FBG CV of less than 5% as the reference, the group with a CV between 5% and 15% and that with a CV over 15% had approximately six-fold and eight-fold increased risks of PC, respectively, and the HRs in all males and in males without diabetes at baseline were almost the same. The linear Cox regression analysis also showed significant associations, with a 2.5% risk increase in all men (HR, 1.025; 95% CI: 1.009–1.042) and a 2.9% risk increase in men without diabetes at baseline (HR, 1.029; 95% CI: 1.011–1.049) with each 1% increase in CV. When setting an FBG range of less than 0.5 mmol/L as the reference, individuals with a range between 0.5 and 1.5 mmol/L and with over 1.5 mmol/L had more than two-fold and three-fold increased risks of PC, respectively, both in all males and in males without diabetes at baseline. A 7.3% increase in the risk of PC was observed in all men (HR, 1.073; 95% CI: 1.021–1.128) and an 8.4% increase in risk of PC was observed in men without diabetes at baseline (HR, 1.084; 95% CI: 1.016–1.157) for every 0.56 mmol/L increase in the range. No significant association was observed between the mean FBG level and PC risk. Sensitivity analyses for associations of the CV and the range of FBG with PC risk showed consistent associations (Appendix A).

Similarly, the RCS results showed that non-linear associations for PC risk were observed with the stability of FBG (as determined by both CV and range), but not with FBG mean (Figure 2). The trends of the curves were similar for all males and for males without diabetes at baseline. The risk of PC increased as the CV and range increased and then stabilized after CV reached approximately 9% and range reached approximately 1 mmol/L.

## 4. Discussion

In this large prospective cohort study of 138,870 men, we evaluated the associations among FBG level, change or stability, and PC risk. We found U-shaped associations between annual FGB changes and PC risk, particularly in men without diabetes at baseline. In addition, FBG instability, represented by a higher CV and larger range, was also associated with an increased risk of PC, whereas single or average FBG levels were not. These results indicate that annual FGB change is better able to predict PC than single or average FBG levels, particularly in men without diabetes. Much attention has been paid to the risks of PC in patients with diabetes in many studies [20,21], and our research suggests that glycemic changes in people without diabetes, both increases and decreases in FBG, are also worth monitoring to identify PC risk.

Our results showed that FBG levels, including baseline FBG, recent FBG, and mean FBG, were not significantly associated with PC risk. Several studies have shown a significant effect of blood glucose on PC risk [13,22,23], while others have reported no significant effect [24]. The Mendelian randomization analysis [25] did not support a causal effect of long-standing diabetes on PC but suggested that PC caused new-onset diabetes, indicating the complex causality between diabetes and PC. Compared with a single FBG level, change in FBG levels had a significantly stronger association with PC in the present study. In 2009, Pannala et al. observed a significant increase in glucose levels before PC diagnosis in a case—control study [15], while associations between groups with different FBG changes and outcomes were not explored. In 2018, Keum et al. [16]. further qualitatively classified the FBG change into four patterns and reported that hyperglycemia might be a predictor of PC; however, this study did not investigate people with decreasing FBG levels. In fact, when identifying FBG changes, both the direction and extent of change should be considered. In addition, those who present FBG changes but who do not meet the diabetes diagnostic criteria should not be overlooked.

Currently, the mechanism by which blood glucose increases positively affect PC development is fairly well understood. Experimental evidence suggests that insulin promotes proliferation and reduces apoptosis in PC cells by increasing the bioavailability of insulin-like growth factor 1 [26,27]. High glucose levels can promote PC cell proliferation [28] and invasion ability [29]. A previous study showed that increased glucose levels could impede adenosine 5′-monophosphate-activated protein kinase (AMPK)-mediated phosphorylation at serine 99, resulting in the destabilization of tet methylcytosine dioxygenase 2 (TET2), followed by the dysregulation of both 5-hydroxymethylcytosine (5hmC) and the tumor suppressive function of TET2 in vitro and in vivo [30].

However, the effect of hypoglycemia on PC remains unclear. A previous study reported that glucose-deprived conditions may protect PC cells from apoptosis [31], which may partly explain the mechanism. In addition, as a form of abnormal glucose metabolism, pathological hypoglycemia is often related to insulinoma, post-bariatric hypoglycemia, and noninsulinoma pancreatogenous hypoglycemia syndrome [32]; the association between these conditions and PC requires further exploration.

As a disease with a poor prognosis, early prevention and intervention are of great importance in improving the survival of patients with PC. Glycemic change combined with other risk factors can help identify groups with high PC risk. At present, PC screening is not recommended for the general population, and accurate and appropriate indicators for identifying high-risk groups are also lacking. Changes in weight, change in blood glucose, and age at diabetes diagnosis were used to construct an enriching model for the new-onset diabetes for pancreatic cancer (END-PDAC) score by Ayush et al. in 2018 using retrospective collected data [9]. Our research provides more detailed risks of PC in groups with different glycemic changes as well as glycemic stability based on perspective data. Although validation in other populations is needed, our findings might be helpful for the prevention and detection of PC and may yield more opportunities for early intervention.

In this study, a major strength was the novelty of concern regarding the change in and stability of FBG, particularly in terms of an FBG decrease, which has been ignored in previous studies. Second, we also focused on individuals without diabetes who were ignored in studies of PC risk. Third, the annual change in FBG levels was quantitatively grouped, which would provide more detailed risk data and would not ignore the risk of those with FBG changes who did not meet diabetes diagnostic criteria. Fourth, a prospective study design with repeated measurements of FBG can minimize recall bias and provide more solid evidence on this association.

However, there were also limitations to the study. First, this study only included Chinese men, thus limiting its generalizability to other populations. In future studies, we will attempt to include more female participants and to validate this association in women. Second, the sample size of participants with diabetes at baseline was limited (less than 1/10th of all males), making it impossible to obtain reliable results among these population. Thus, this part of the analysis was not included in the main analysis. Third, the last FBG measurement of a small number of participants who were later diagnosed with PC were measured within 1 year of PC diagnosis, which might cause confusion surrounding the cause and effect. Therefore, we mainly emphasized the suggestive role of FBG changes before the diagnosis of PC, rather than its etiology. Fourth, some covariates had missing values, despite our efforts, which may have a slight impact on the results of the multivariate analysis.

## 5. Conclusions

In summary, this population-based cohort study demonstrated that glycemic change and stability are associated with a risk of PC. In individuals with significant changes in FBG over time, a markedly higher risk of PC was observed than in those with relatively stable FBG levels. For the prevention and early detection of PC, attention should be paid not only to people with increasing FBG levels but also to those with decreasing FBG levels.

## Figures and Tables

**Figure 1 cancers-14-03403-f001:**
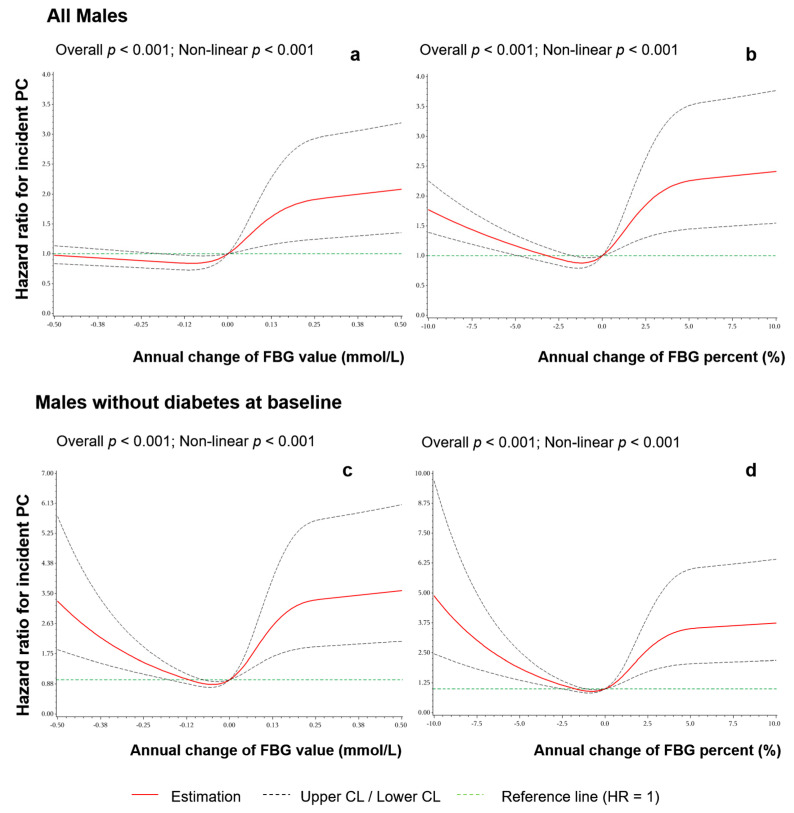
Non-linear associations between annual FBG change and the risk of PC. (**a**) Annual change in FBG value in all males, (**b**) annual change in FBG percent in all males, (**c**) annual change in FBG value in males without diabetes at baseline, and (**d**) annual change in FBG percent in males without diabetes at baseline.

**Figure 2 cancers-14-03403-f002:**
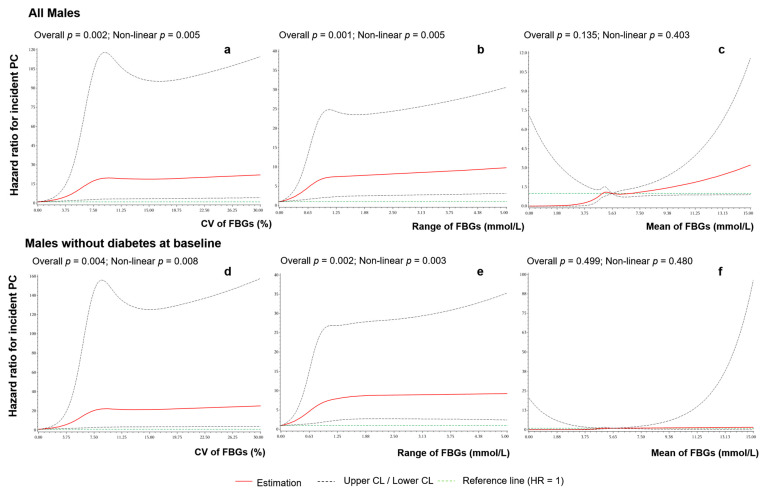
Non-linear associations between FBG stability and the risk of PC. (**a**) CV of FBGs in all males, (**b**) range of FBGs in all males, (**c**) mean of FBGs in all males, (**d**) CV of FBGs in males without diabetes, (**e**) range of FBGs in males without diabetes, and (**f**) mean of FBGs in males without diabetes.

**Table 1 cancers-14-03403-t001:** Characteristics of participants included in the analysis.

Variables	Total	Times of FBG Measurements			*p*
		1	2~3	4~6	
No. of participants	138,870	32,238	44,479	62,153	
No. of incident PC	135	42	68	25	
Follow-up time, y	11.39 (7.03,13.06)	6.24 (3.91,10.14)	9.2 (6.53,12.76)	12.99 (12.51,13.18)	<0.001
Duration of FBG measurements	6.26 (1.67,10.1)	-	4.15 (2.36,6.13)	10.2 (8.24,10.85)	<0.001
Age(y)					
M (P25, P75)	51 (39,59)	53 (36,63)	51 (37,60)	50 (41,56)	<0.001
Age < 40, No. (%)	35,424 (25.51)	9417 (29.21)	12,530 (28.17)	13,477 (21.68)	<0.001
40 ≤ Age < 60, No. (%)	70,763 (50.96)	11,831 (36.70)	20,450 (45.98)	38,482 (61.91)	
60 ≤ Age < 80, No. (%)	30,003 (21.61)	9427 (29.24)	10,551 (23.72)	10,025 (16.13)	
≥80, No. (%)	2680 (1.93)	1563 (4.85)	948 (2.13)	169 (0.27)	
BMI (kg/m^2^)					
M (P25,P75)	24.86 (22.71,27.12)	24.77 (22.49,26.99)	24.77 (22.53,26.99)	25.04 (22.89,27.34)	<0.001
BMI<18.5	2362 (1.7)	704 (2.18)	835 (1.88)	823 (1.32)	<0.001
18.5≤BMI<25.0	69,563 (50.09)	16,884 (52.37)	23,070 (51.87)	29,609 (47.64)	
25.0≤BMI<30.0	56,551 (40.72)	12,364 (38.35)	17,371 (39.05)	26,816 (43.15)	
BMI≥30.0	10,394 (7.48)	2286 (7.09)	3203 (7.20)	4905 (7.89)	
Education status, No. (%)					
Junior middle school or lower education	96,143 (69.23)	16,908 (52.45)	31,684 (71.23)	47,551 (76.51)	<0.001
Senior middle school	19,028 (13.7)	2808 (8.71)	6445 (14.49)	9775 (15.73)	
College and above	11,652 (8.39)	2391 (7.42)	4439 (9.98)	4822 (7.76)	
Missing	12,047 (8.68)	10,131 (31.43)	1911 (4.30)	5 (0.01)	
Smoking history, No. (%)					
Never smoker	66,660 (48)	14,633 (45.39)	22,567 (50.74)	29,460 (47.40)	<0.001
Ever-smoker	66,945 (48.21)	12,686 (39.35)	21,567 (48.49)	32,692 (52.60)	
Missing	5265 (3.79)	4919 (15.26)	345 (0.78)	1 (0.00)	
Drinking history, No. (%)					
Never drinker	66,662 (48)	13,934 (43.22)	23,878 (53.68)	28,850 (46.42)	<0.001
Ever drinker	63,020 (45.38)	9654 (29.95)	20,064 (45.11)	33,302 (53.58)	
Missing	9188 (6.62)	8650 (26.83)	537 (1.21)	1 (0.00)	
Anti-diabetic medications use history, No. (%)					
No	118,793 (85.54)	21,875 (67.85)	37,157 (83.54)	59,761 (96.15)	<0.001
Yes	2755 (1.98)	689 (2.14)	910 (2.05)	1156 (1.86)	
Missing	17,322 (12.47)	9674 (30.01)	6412 (14.42)	1236 (1.99)	
Working environment, No. (%)					
Above the ground	76,876 (55.36)	15,178 (47.08)	25,134 (56.51)	36,564 (58.83)	<0.001
Underground without dust exposure	13,138 (9.46)	2160 (6.70)	4409 (9.91)	6569 (10.57)	
Underground with dust exposure	38,965 (28.06)	5877 (18.23)	14,068 (31.63)	19,020 (30.60)	
Missing	9891 (7.12)	9023 (27.99)	868 (1.95)	0 (0.00)	

FBG: fasting blood glucose; PC: pancreatic cancer; BMI: body mass index.

**Table 2 cancers-14-03403-t002:** Hazard ratios (HRs) for the association between FBG annual change and pancreatic cancer risk.

FBG Annual Change		All Males			Males without Diabetes at Baseline		
		P-Ys (Case)	HR (95% CI) ^a^	Adjusted HR (95% CI) ^b^	P-Ys (Case)	HR (95% CI) ^a^	Adjusted HR (95% CI) ^b^
Value (mmol/L) per year	−0.05~0.05	384,317.27 (10)	Ref	Ref	372,674.58 (9)	Ref	Ref
	<−0.05	248,300.50 (26)	4.247 (2.047, 8.810) **	4.010 (1.920, 8.375) **	199,506.99 (19)	4.161 (1.882, 9.200) **	3.976 (1.797, 8.798) **
	0.05~0.15	298,390.94 (15)	1.940 (0.872, 4.318)	1.816 (0.815, 4.045)	289,282.46 (15)	2.154 (0.942, 4.921)	2.022 (0.884, 4.623)
	≥0.15	253,207.62 (42)	6.702 (3.362, 13.361) **	5.897 (2.935, 11.848) **	221,647.65 (38)	7.436 (3.595, 15.383) **	6.769 (3.254, 14.080) **
Percent (%) per year	−1%~1%	400,348.79 (11)	Ref	Ref	303,446.40 (9)	Ref	Ref
	<−1%	233,833.54 (25)	4.121 (2.027, 8.379) **	3.851 (1.883, 7.878) **	267,307.19 (19)	2.498 (1.130, 5.523) *	2.414 (1.092, 5.340) *
	1%~3%	290,858.88 (14)	1.761 (0.799, 3.879)	1.671 (0.758, 3.682)	278,220.88 (13)	1.582 (0.676, 3.702)	1.502 (0.642, 3.516)
	≥3%	259,175.13 (43)	6.332 (3.264, 12.282) **	5.512 (2.827, 10.748) **	234,137.22 (40)	6.012 (2.917, 12.394) **	5.390 (2.603, 11.162) **

FBG, fasting blood glucose; P-Ys, person years; HR, hazard ratio; BMI, body mass index. ^a^: univariate analysis; ^b^: adjusted for age, education level, smoking history, drinking history, BMI, anti-diabetes drug use, and BMI change; * *p* < 0.05; ** *p* < 0.01.

**Table 3 cancers-14-03403-t003:** Hazard ratios (HRs) for the association between CV of FBGs and pancreatic cancer risk.

FBG Stability		All Males					Males without Diabetes at Baseline				
		P-Ys (Case)	HR (95% CI) ^a^	Adjusted HR (95% CI) ^b^	Linear HR (95% CI) ^c^	*P _for trend_*	P-Ys (Case)	HR (95% CI) ^a^	Adjusted HR (95% CI) ^b^	Linear HR (95% CI) ^c^	*P _for trend_*
CV (%)	<5	187,842.94 (4)	Ref	Ref	1.025 (1.009, 1.042) **	0.003	182,445.65 (4)	Ref	Ref	1.029 (1.011, 1.049) **	0.002
	5~15	732,002.46 (55)	3.317 (1.202, 9.156) *	6.039 (2.173, 16.783) **			701,661.54 (52)	3.203 (1.158, 8.859) *	5.918 (2.123, 16.493) **		
	≥15	264,374.86 (34)	5.663 (2.009, 15.967) **	8.039 (2.815, 22.958) **			199,004.5 (25)	5.414 (1.883, 15.563) **	7.939 (2.747, 22.947) **		
Range (mmol/L)	<0.5	164,305.43 (10)	Ref	Ref	1.073 (1.021, 1.128) **	0.005	160,739.67 (10)	Ref	Ref	1.084 (1.016, 1.157) *	0.015
	0.5~1.5	617,668.22 (48)	1.167 (0.590, 2.308)	2.63 (1.314, 5.265) **			604,832.99 (46)	1.127 (0.568, 2.236)	2.621 (1.304, 5.269) **		
	≥1.5	402,246.6 (35)	1.274 (0.630, 2.578)	3.553 (1.671, 7.555)**			317,539.03 (25)	1.138 (0.545, 2.375)	3.773 (1.74, 8.181) **		
Mean (mmol/L)	<5.6	775,037.24 (56)	Ref	Ref	1.074 (0.998, 1.157)	0.058	771,836.29 (56)	Ref	Ref	1.084 (0.939, 1.252)	0.272
	5.6~7	284,942.69 (25)	1.209 (0.755, 1.938)	1.085 (0.668, 1.763)			265,903.49 (22)	1.136 (0.694, 1.860)	1.090 (0.662, 1.794)		
	≥7	124,240.33 (12)	1.327 (0.711, 2.475)	0.854 (0.354, 2.058)			45,371.91 (3)	0.892 (0.279, 2.848)	1.151 (0.342, 3.88)		

FBG, fasting blood glucose; P-Ys, person years; HR, hazard ratio; BMI, body mass index. ^a^: univariate analysis; ^b^: adjusted for age, education level, smoking history, drinking history, baseline BMI, BMI change, anti-diabetes drug use, times of FBG tests, and duration of FBG tests; ^c^: linear effect per 1 increase in CV or 0.56 mmol/L (10mg/dL) increase in range and mean, and adjusted for age, education level, smoking history, drinking history, baseline BMI, BMI change, anti-diabetes drug use, times of FBG tests, and duration of FBG tests; * *p* < 0.05; ** *p* < 0.01.

## Data Availability

The data that support the findings of this study are available from the corresponding author upon reasonable request and approval from Kailuan General Hospital.

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
