# Peer review of "Association between Temporal Glycemic Change and Risk of Pancreatic Cancer in Men: A Prospective Cohort Study"

_cancers, 2022, doi:10.3390/cancers14143403_

Round 1

Reviewer 1 Report

In this work, the authors correlate pancreatic cancer risk to fasting blood glucose in a population of Chinese man with and without diabetes.  Studies are still needed in women and other populations but the authors lay groundbreaking findings in this field.  The review recommends this work for publication with a few suggestions.

1.       The authors discuss measuring change in fasting blood glucose using first and last measurements for each subject, but have they considered using all measurements and applying a generalized estimating equation?

2.       The authors mention that very few women were available for analysis but a discussion of acquiring more women in future studies might be helpful.

3.       The authors might want to elaborate on the reason for including missing indicators in the analyses further.

4.       Transformation of fasting blood glucose levels (e.g., log transformation) may help improve stability.

Reviewer 2 Report

The association between glycemic change and PC risk has been studied for many years. The hyperglycemia, which is caused by insulin resistance and inability to suppress inappropriate hepatic glucose release, has been shown to enhance proliferation, promotes epithelial–mesenchymal transition and cancer stem cells’ properties, and metastatic potential in pancreatic cancer. There is a 1.5- to 2.0-fold increase in the risk of developing pancreatic cancer in type 2 diabetes mellitus patients. In the manuscript by Jie cai et al, the authors analyzed the association between glycemic change and PC risk using a cohort from Kailuan Group in the city of Tangshan since 2006. A total of 138,870 males with available fasting blood glucose (FBG) levels, including 106,632 males with at least two FBG measurements, were analyzed in this study. The authors observed a U-shape associations between the annual FBG change and PC risk. And the participants with annual FBS change showed a higher risk for PC. It was concluded temporal FBG changes may be an important factor for identifying populations with high PC risk. Here are the major concerns: 1. For all males in Figure 1, decreased annual FBG values and FBG percentage showed different hazard ratio for incident PC. Please discuss the possible reasons. 2. It is quite interesting to see the association between the decreased FBG and PC. Is the decreased FBG physiological or pathological in the participants? Here is the minor concern: Line 16: redundant space in this line.
